# On-Shore Plastic Waste Detection with YOLOv5 and RGB-Near-Infrared Fusion: A State-of-the-Art Solution for Accurate and Efficient Environmental Monitoring

Owen Tamin [1], Ervin Gubin Moung [1,2,*], Jamal Ahmad Dargham [3], Farashazillah Yahya [1], Ali Farzamnia [3], Florence Sia [1], Nur Faraha Mohd Naim [1] and Lorita Angeline [3]

[1]  Faculty of Computing and Informatics, Universiti Malaysia Sabah, Kota Kinabalu 88400, Sabah, Malaysia; owentamin1996@gmail.com (O.T.); fara.yahya@ums.edu.my (F.Y.); florence.sfs@ums.edu.my (F.S.); faraha.naim@ums.edu.my (N.F.M.N.)
[2]  Data Technologies and Applications (DaTA) Research Group, Universiti Malaysia Sabah, Kota Kinabalu 88400, Sabah, Malaysia
[3]  Faculty of Engineering, Universiti Malaysia Sabah, Kota Kinabalu 88400, Sabah, Malaysia; jamalad@ums.edu.my (J.A.D.); alifarzamnia@ums.edu.my (A.F.); angeline.lorita@ums.edu.my (L.A.)
*  Correspondence: ervin@ums.edu.my

**Abstract:** Plastic waste is a growing environmental concern that poses a significant threat to onshore ecosystems, human health, and wildlife. The accumulation of plastic waste in oceans has reached a staggering estimate of over eight million tons annually, leading to hazardous outcomes in marine life and the food chain. Plastic waste is prevalent in urban areas, posing risks to animals that may ingest it or become entangled in it, and negatively impacting the economy and tourism industry. Effective plastic waste management requires a comprehensive approach that includes reducing consumption, promoting recycling, and developing innovative technologies such as automated plastic detection systems. The development of accurate and efficient plastic detection methods is therefore essential for effective waste management. To address this challenge, machine learning techniques such as the YOLOv5 model have emerged as promising tools for developing automated plastic detection systems. Furthermore, there is a need to study both visible light (RGB) and near-infrared (RGNIR) as part of plastic waste detection due to the unique properties of plastic waste in different environmental settings. To this end, two plastic waste datasets, comprising RGB and RGNIR images, were utilized to train the proposed model, YOLOv5m. The performance of the model was then evaluated using a 10-fold cross-validation method on both datasets. The experiment was extended by adding background images into the training dataset to reduce false positives. An additional experiment was carried out to fuse both the RGB and RGNIR datasets. A performance-metric score called the Weighted Metric Score (WMS) was proposed, where the WMS equaled the sum of the mean average precision at the intersection over union (IoU) threshold of 0.5 (mAP@0.5) × 0.1 and the mean average precision averaged over different IoU thresholds ranging from 0.5 to 0.95 (mAP@0.5:0.95) × 0.9. In addition, a 10-fold cross-validation procedure was implemented. Based on the results, the proposed model achieved the best performance using the fusion of the RGB and RGNIR datasets when evaluated on the testing dataset with a mean of mAP@0.5, mAP@0.5:0.95, and a WMS of 92.96% ± 2.63%, 69.47% ± 3.11%, and 71.82% ± 3.04%, respectively. These findings indicate that utilizing both normal visible light and the near-infrared spectrum as feature representations in machine learning could lead to improved performance in plastic waste detection. This opens new opportunities in the development of automated plastic detection systems for use in fields such as automation, environmental management, and resource management.

**Keywords:** plastic waste detection; environmental impact; object detection; YOLOv5; machine learning; deep learning; RGB-NIR feature representation; near-infrared; data processing; image feature learning; automated plastic detection

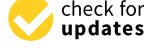

## 1. Introduction

Plastic waste refers to the accumulation of plastic objects in the environment that pose a threat to humans, wildlife, and their habitats. In other words, it is the massive number of plastic objects that were neglected and not recycled, which ended up in landfills. Additionally, plastic objects were discovered on the ocean's surface. It was estimated that around 23 million tons of plastic objects had entered the world's oceans [1]. Human actions were mainly responsible for plastic waste due to poor management and attitude. According to recent studies, the global rate of plastic recycling was alarmingly low, with only 18% of plastic objects being recycled and a concerning 24% being incinerated [2]. This highlights the urgent need for more effective and sustainable approaches to managing plastic waste. Therefore, there is a need to develop a plastic detection system to resolve the plastic waste issue.

A plastic detection system consists of locating and detecting plastic objects in an area effectively and efficiently. One potential imaging technique that can be used for this purpose is near-infrared spectroscopy (NIRS). NIRS is a method that gathers near-infrared information about the electromagnetic spectrum in an object. The method is fast, accurate, and safe due to its non-contact and non-destructive properties [3]. Wu et al. and Rani et al. stated that the spectrometer detects plastics in the near-infrared (NIR) spectrum of 900 to 1700 nanometers (nm) [4,5]. On the other hand, Moshtaghi et al. observed that plastic could be detected in a spectrum as low as 850 to 900 nm with small plastic reflectance information [6].

However, NIRS techniques face a few limitations in detecting plastic objects. One of the main limitations is the inability to detect black plastics. It was found that the incorporation of a small quantity of carbon black in plastic results in the absorption of all light within the near-infrared spectral region. The light absorption causes the NIR spectrometer to be unable to record and detect black plastic in the system [7,8]. Another limitation is the accessibility of the NIR technology. NIR technology requires an expensive and complex setup. Cameras are frequently integrated into complex systems to optimize the illumination of targeted objects through the utilization of light. This approach allows for a more comprehensive and efficient collection of data and information.

In addition to NIRS techniques, machine learning (ML) is an alternative form of a plastic detection system. ML was extensively used as a data processing technique for solving food recognition [9], facial expression and face recognition [10,11], and disease identification [12]. The method utilized and developed a computer system capable of learning and analyzing through algorithms and data. The incorporation of ML techniques by Bobulski and Kubanek in a sorting system resulted in the successful implementation of automatic plastic detection [13]. The proposed system achieved an accuracy of 97.43% with an image size of $120 \times 120$ for 2000 plastic waste images.

Object detection is an ML discipline that involves finding and classifying objects within specific categories. This often involves drawing boxes around the objects and labeling them with their corresponding class. You Only Look Once (YOLO), a state-of-the-art object detection method, gained significant attention in recent years due to its reliability, speed, and accuracy [14]. This innovative approach demonstrated significant potential in a variety of applications, with its effectiveness being consistently demonstrated in numerous studies and practical implementations. YOLO was based on a convolutional neural network (CNN) that uses a single neural network. YOLO utilizes a single forward propagation process to simultaneously predict multiple bounding boxes and assign class probabilities for each box within an image. This unique approach allows for efficient and accurate object detection and contributes to the success and popularity of the method. It is worth noting that the effectiveness of this approach was consistently demonstrated in a variety of applications, making YOLO a highly valuable tool in the field. Currently, there are five versions of YOLO, including YOLOv1 [14], YOLOv2 [15], YOLOv3 [16], YOLOv4 [17], and YOLOv5 [18]. YOLOv1 and YOLOv2 were ineffective in detecting small targets in the images. Therefore, multi-scale detection was added to YOLOv3 and its subsequent

version for better detection of tiny targets. However, YOLOv3 only adopted a single anchor point in their architecture responsible for multiple ground truths, and this problem led to slower and more inaccurate object detection. Therefore, YOLOv4 was improved by adopting several anchor points for ground truth detection, leading to faster and more accurate performance than YOLOv3 [18]. On the other hand, YOLOv5 implemented the Pytorch framework in its architecture. The advantage of the Pytorch framework over the Darknet used by YOLOv4 is its user-friendly interface for training the model and producing better real-time results.

This paper uses YOLOv5 to identify plastic waste in two different datasets. The first dataset consists of images captured in the RGB color space, while the second dataset consists of images captured in the RGNIR color space. This paper extends the work that was conducted in [19–21]. Since NIR spectrometers were able to detect plastic as low as 850 nm, this paper aims to observe if ML could detect plastic better in NIR images [6]. This paper presents two notable contributions to the field as follows:

1. Two datasets of plastic waste were established, one comprising the RGB color space, and the other comprising their corresponding RGNIR color space. Additionally, a dataset of background images devoid of plastic waste in both the RGB and RGNIR color spaces was compiled.

2. The present research provides a systematic comparison of the performance of two color spaces, RGB and RGNIR, when used in a YOLOv5-based system for detecting plastic waste. Our study offers a comprehensive evaluation of the effectiveness of these color spaces in this context. The analysis aims to provide insight into the effectiveness of these two approaches in detecting plastic waste and to identify any potential advantages or disadvantages of each method. Through this comparison, we hope to contribute to the understanding of the role of color space in the effectiveness of object detection systems.

This paper is divided into seven sections. Section 1 covers the research background, motivation, and contribution. Section 2 describes the related work of this study. Section 3 explains the proposed method, while Section 4 elaborates on the dataset preparation. Section 5 presents the experiment used to conduct this study. The results and discussion are explained in Section 6. Section 7 presents the main conclusions drawn from the analysis. Additionally, Section 8 discusses future directions for research, highlighting potential areas for further investigation and development.

## 2. Related Works

Recently, more studies have used YOLOv5 as part of object detection models to identify waste. Córdova et al. conducted several experiments to test the performance of YOLOv5, RetinaNet, EfficientDet, Faster R-CNN, and Mask R-CNN on litter detection based on the TACO dataset [22]. The TACO dataset consisted of 1500 images with image sizes ranging from 842 × 474 to 6000 × 4000 pixels of 60 litter categories involving 4784 annotations [23]. The results of this study demonstrate that the YOLOv5 algorithm outperformed all four state-of-the-art CNN architectures in terms of both speed and accuracy of litter detection. This finding is significant as it highlights the effectiveness of YOLOv5 in detecting litter and suggests that it may be a valuable tool in litter detection applications. It is worth noting that the ability to detect litter accurately and efficiently is crucial in addressing the negative impacts of litter on the environment and human health. A similar study was conducted in the past, in which the performance of various YOLO models for litter detection was evaluated using the same dataset [24]. According to the results presented in the previously mentioned study [24], the YOLOv5 model demonstrated superior performance compared to the YOLOv3 model, with a detection speed of 5.52 frames per second and a mean average precision of 97.62%. These findings suggested that YOLOv5 may be a particularly effective algorithm for litter detection, with its high speed and accuracy making it a valuable tool in this application.

Wu et al. utilized GC-YOLOv5, a YOLOv5 object detection network, to build a garbage classification model. The YOLOv5s pre-training weight was used during training [25]. The proposed model in [25] demonstrated promising results, with accuracy, recall, mean average precision at the intersection over union (IoU) threshold of 0.5 (mAP@0.5), and mean average precision averaged over different IoU thresholds ranging from 0.5 to 0.95 (mAP@0.5:0.95) values of 99.86%, 100%, 99.59%, and 64.70%, respectively. These findings suggested that the model was highly effective in accurately detecting and identifying objects with high recall and accuracy rates. Additionally, the mAP@0.5 and mAP@0.5:0.95 values indicated that the model was able to accurately detect a high percentage of objects at various levels of overlap. A similar study utilized the YOLOv5s architecture to build garbage classification and detection models in rural areas [26]. The proposed model built a better background network structure by adding an attention combination mechanism. The mechanism described in [26] involved the combination of both channel and spatial dimensions as feature information, resulting in a more comprehensive representation of the proposed model. This approach allowed for the incorporation of a greater range of information and detail, potentially enhancing the accuracy and effectiveness of the model in various object detection applications. The accuracy, recall, and mAP@0.5 were reported as 93.5%, 91.1%, and 96.4%, respectively.

A previous study sought to improve the YOLOv5s architecture by introducing a feature map attention (FMA) at the end of the backbone layer [27]. This modification represented a significant advancement in the field, as it allowed for the incorporation of additional information and detail, potentially enhancing the accuracy and effectiveness of the model in various object detection applications. FMA improved the feature extraction ability of the proposed model in detecting eight waste categories of floating objects. The model merged the labeled target objects with the background images of the clean river to create real-scene environment settings. Based on the results, the model obtained the mAP@0.5 of 79.41% on the testing dataset. Besides YOLOv5s, a study improved the feature extraction ability of YOLOv5m by adding a supervised attention mechanism [28]. The model was combined with a multimodal knowledge graph to improve garbage detection from images and videos in a real scene. Due to the combination, the model achieved mAP@0.5 of 72.8% compared to the original YOLOv5m with a performance of 72.8%. Table 1 presents the summary of the related work.

**Table 1.** Summary of related work.

| Authors | Models Used | Dataset | Performance Metrics | Findings |
|---|---|---|---|---|
| Córdova et al., 2022 [22] | YOLOv5, RetinaNet, EfficientDet, Faster R-CNN, Mask R-CNN | TACO | Speed, Accuracy | YOLOv5 outperformed other CNN architectures in terms of speed and accuracy in detecting litter, suggesting it may be a valuable tool in litter detection applications. |
| Proença and Simões, 2020 [23] | Unspecified | TACO | Unspecified | The TACO dataset consisted of 1500 images with image sizes ranging from 842 × 474 to 6000 × 4000 pixels of 60 litter categories involving 4784 annotations. |
| Lv et al., 2022 [24] | YOLOv5 | TACO | Detection speed, mAP | YOLOv5 had superior performance compared to YOLOv3, with a detection speed of 5.52 frames per second and a mean average precision of 97.62%. |
| Wu et al., 2021 [25] | GC-YOLOv5 | Unspecified | Accuracy, Recall, mAP@0.5 | GC-YOLOv5 demonstrated high accuracy and recall rates, and mAP@0.5 values indicated accurate detection of a high percentage of objects with varying levels of overlap. |

**Table 1.** *Cont.*

| Authors | Models Used | Dataset | Performance Metrics | Findings |
|---|---|---|---|---|
| Jiang et al., [26] | YOLOv5s | Unspecified | Accuracy, Recall, mAP@0.5 | YOLOv5s-based garbage classification and detection model in rural areas improved background network structure by adding an attention combination mechanism, leading to better feature representation and potentially enhancing accuracy and effectiveness of the model. The model achieved high accuracy, recall, and mAP@0.5 rates. |
| Lin et al., 2021 [27] | YOLOv5s | Floating objects | mAP@0.5 | The addition of feature map attention (FMA) at the end of the backbone layer improved feature extraction ability in detecting floating waste categories, and the model achieved an mAP@0.5 of 79.41% on the testing dataset. |
| Zang et al., 2022 [28] | YOLOv5m | Images, Videos | mAP@0.5 | YOLOv5m improved feature extraction ability by adding a supervised attention mechanism combined with a multimodal knowledge graph to improve garbage detection in real-scene images and videos, achieving an mAP@0.5 of 72.8%. |

## 3. Proposed Method

This section describes the research methodology of this study.

### 3.1. Research Overview

The proposed approach for developing a plastic waste detection model is depicted in Figure 1. The proposed flowchart outlines a comprehensive approach for developing a plastic waste detection model. The process begins with image acquisition, followed by data pre-processing, which involves several steps such as data cleaning, cropping, resizing, and annotation. The pre-processed data is then split into three datasets including training, validation, and testing, to ensure the model's generalizability. The training phase involves training and validating the model using the training and validation datasets, respectively. The fine-tuned plastic waste detection model is then evaluated on the testing dataset to assess its performance on an unseen dataset. The evaluation process includes assessing the model's recall rate, precision rate, and mAP@0.5. This proposed methodology offers a systematic and effective approach for developing a plastic waste detection model, which could have significant implications for managing plastic waste in the environment.

### 3.2. Object Detection Model

The YOLOv5 model is a state-of-the-art object detection model that utilizes a combination of three components, including (i) CSPDarknet53 as the backbone, (ii) Path Aggregation Network (PANet) as the neck, and (iii) three YOLO heads [29]. The model is implemented using PyTorch as the framework, and the open-source code is publicly available on GitHub. The combination of these components allows for the effective and efficient detection of objects, making YOLOv5 a valuable tool in various object detection applications. In the YOLOv5 object detection model, the image is first processed by the CSPDarknet53 component for feature extraction, which utilizes cross-stage partial networks. This extracted feature representation is then passed to the PANet component, which contains a feature pyramid network (FPN). The PANet component of the YOLOv5 object detection model consists of multiple layers that facilitate the propagation of low-level features through the use of both top-down and bottom-up connections. This design allows for the efficient and

effective processing of the feature representation extracted by the CSPDarknet53 component. The resulting feature representation is then passed to the three heads layer, which generates predictions from anchor boxes for the purpose of object detection. Figure 2 shows the network architecture of YOLOv5.

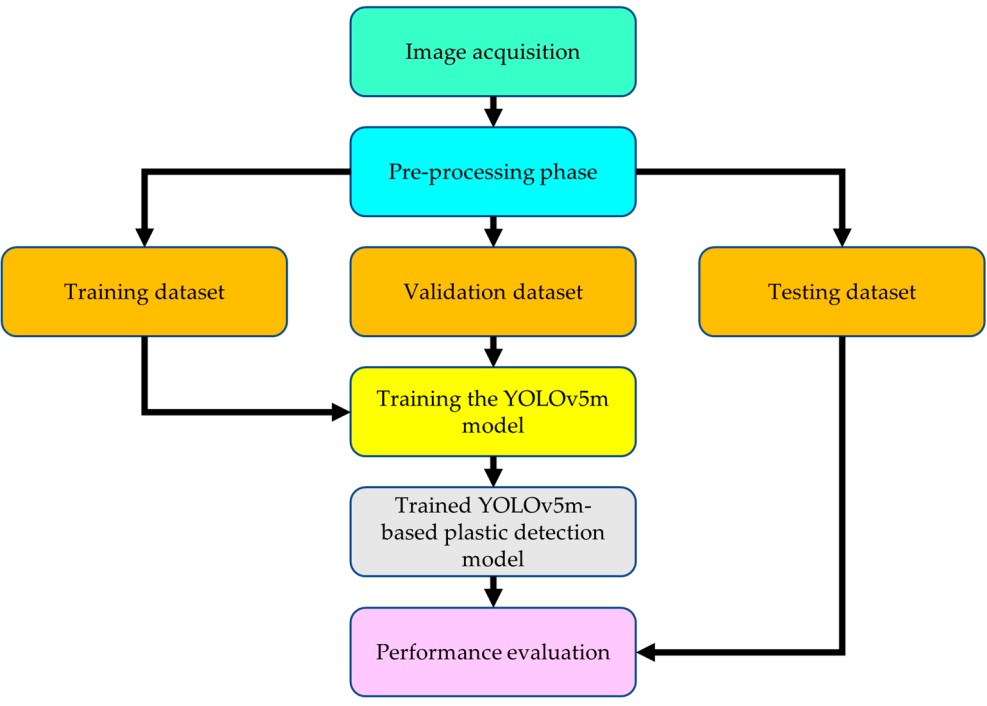

**Figure 1.** Flowchart of the proposed method.

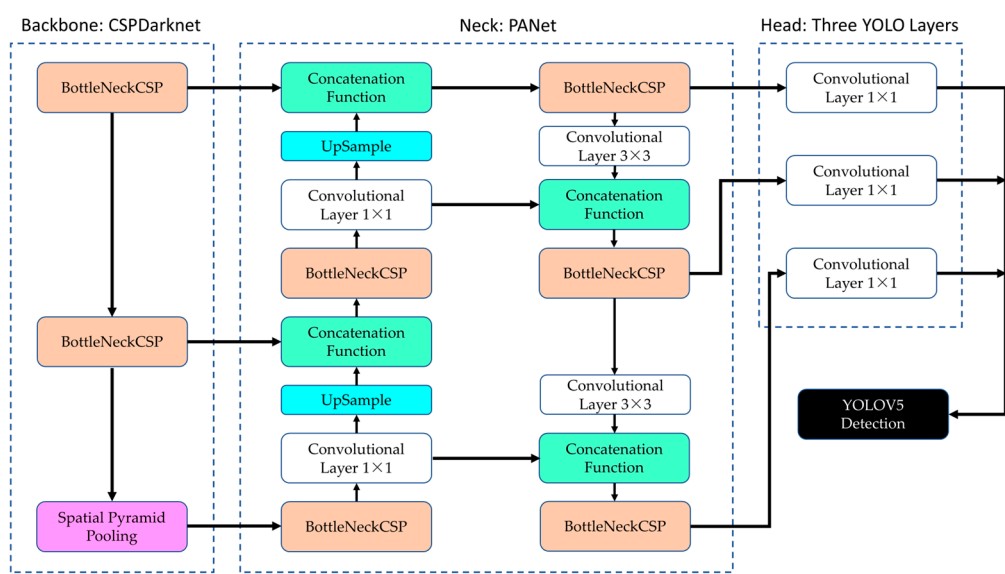

**Figure 2.** YOLOv5 network architecture.

The YOLOv5 object detection model utilizes an integrated focus layer within its backbone, which helps to reduce the memory requirements of the CUDA platform. This optimization allows for higher accuracy of the model to be achieved in a shorter amount of time, making it a valuable tool in various object detection applications. Furthermore, YOLOv5 is available in five different sizes, including (i) extra-large (YOLOv5x), (ii) large (YOLOv5l), (iii) medium (YOLOv5m), (iv) small (YOLOv5s), and (v) nano (YOLOv5n) [30]. This range of model sizes allows for the optimization of the model for different applications and computing environments, making it a highly flexible and adaptable tool in the field.

The different model sizes available within the YOLOv5 object detection algorithm offers varying trade-offs in terms of speed, accuracy, and complexity. Smaller models are generally faster but may lack accuracy in detecting smaller objects. Conversely, larger models consume more computational resources but tend to perform better in detecting large and complex images.

Previous studies have demonstrated that YOLOv5 outperforms other state-of-the-art object detection algorithms, including RetinaNet, EfficientDet, Faster R-CNN, and Mask R-CNN, in terms of both speed and accuracy [22]. In this study, the YOLOv5m model is specifically chosen for its high accuracy, speed, and balanced complexity, falling in the middle of the range of the available model sizes. This selection is based on the demonstrated superiority of YOLOv5 in previous studies and its ability to optimize performance for different computing environments.

### 3.3. Performance Metrics

This study utilizes three primary performance metrics, including precision, recall, and mean average precision (mAP). These metrics allow for the quantification of the model's ability to accurately and reliably detect objects and are essential in evaluating the suitability of the model for various object detection tasks. Two variations of the mAP metric were also used, (i) mAP@0.5 and (ii) mAP@0.5:0.95. The mAP is a measure of the average precision (AP) calculated for all classes. As plastic waste is the only class being considered in this study, the computed mAP represented the AP for all plastics. The intersection over union (IoU) had to be specified first to calculate the AP. The IoU is a metric that quantifies the overlap between the ground truth bounding box and the predicted bounding box. It is calculated as the ratio of the intersection area to the union area of the two bounding boxes. This metric is widely used in the evaluation of object detection models, as it allows for the assessment of the model's ability to accurately identify and locate objects within an image. An example of the IoU is shown in Figure 3. To classify a detection as correct or incorrect, a threshold value is compared to the IoU value calculated for the predicted and ground truth bounding boxes. A threshold value of 0.5 is commonly used in the evaluation of object detection models and is the value chosen in this study.

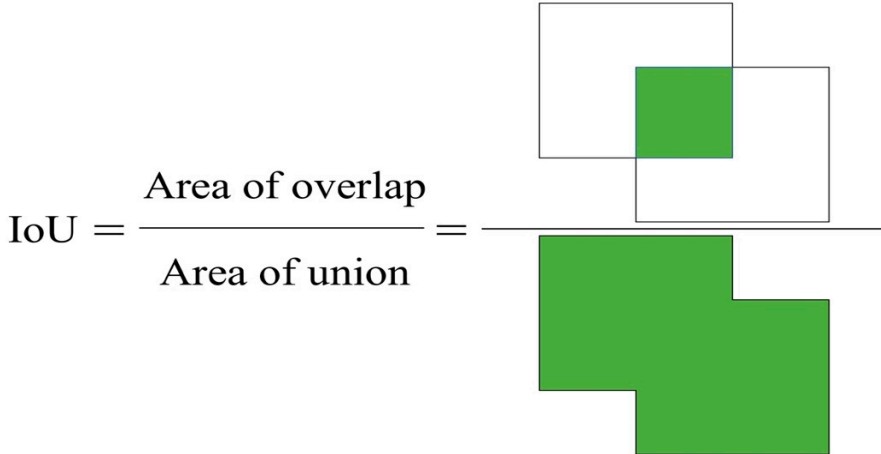

$$IoU = \frac{\text{Area of overlap}}{\text{Area of union}} =$$

**Figure 3.** Example of intersection over union.

To appropriately evaluate the performance of an object detection model, the relationship between the IoU value and the threshold value has to be considered. The following terminologies are introduced specifically to describe this relationship: True Positive (TP) for IoU $\geq 0.5$, False Positive (FP) for IoU $< 0.5$, and False Negative (FN). TP indicates a correct detection of a ground truth bounding box, whereas FP indicates an inaccurate detection of an item. FN indicates that the model is unable to detect the presence of an object within the image. True Negative (TN) was not factored into the performance calculation because TN indicates the absence of any targeted objects in the detected image [31].

While TNs are an important aspect of a model's performance, they are typically not included in the mAP calculation for object detection because mAP is specifically designed to evaluate the model's ability to detect objects and not its ability to correctly identify the absence of an object. As all plastic objects in the images have already been labeled during the pre-processing stage, TN is not relevant in the context of this work. Thus, the evaluation of the model's performance is based solely on the terms TP, FP, and FN. These terms are used to calculate the Precision (P) and Recall (R) metrics, which provide insight into the accuracy and reliability of the model in detecting objects within an image. P represents the model's ability to accurately identify relevant objects and is calculated as the proportion of correct predictions. R, on the other hand, represents the model's ability to locate all relevant ground truth bounding boxes and is calculated as the ratio of positive predictions that are correct from all existing ground truths. The formulas for P and R are defined in Equations (1) and (2), respectively.

$$P = \frac{TP}{TP + FP} \tag{1}$$

$$R = \frac{TP}{TP + FN} \tag{2}$$

To visualize the trade-off between P and R for different thresholds, a Precision–Recall curve was constructed. This curve allows for the identification of the optimum threshold that maximizes both metrics and is a valuable tool in optimizing the performance of an object detection model. Then, the AP metric is computed as a summary of the Precision–Recall curve, indicating the mean of all precisions. The AP is derived as the weighted mean of precisions at each threshold, with the weight reflecting the increase in recall relative to the prior threshold. The formula for the AP is given by Equation (3).

$$AP = \sum^{k=i-1} [R(k) - R(k+1)] \times P(k) \tag{3}$$

where
  i = number of thresholds.
  The mAP is then determined by averaging the AP across all classes, as demonstrated by Equation (4).

$$mAP = \frac{1}{n} \sum^{k=n} A_k \tag{4}$$

where
  n = number of classes;
  $A_k$ = the AP of class k.
  In summary, the calculation of mAP in this study is focused on a singular class, plastic waste, and two variations of the metric are obtained from the general formula for mAP, namely, mAP@0.5 and mAP@0.5:0.95.

## 4. Dataset

This section details the preparation of the datasets for this study.

### 4.1. Data Acquisition

Three public places in Kota Kinabalu, Sabah, Malaysia were identified as having a high concentration of plastic garbage in a natural environment for the purpose of data collection; these places include Moyog Riverbank, Tanjung Lipat Beach, and Tanjung Aru Beach. To create two datasets, images of plastic waste were acquired using two different types of cameras at these locations, with the only criteria being that the images are taken during the day, regardless of the weather condition.

It should be noted that the daylight conditions, such as sunny or cloudy weather, were not taken into account during the data collection process. While daylight conditions can

affect the visibility and clarity of the images, the decision not to consider this factor was made to ensure that the dataset is inclusive of diverse environmental conditions. By doing so, the resulting dataset is more comprehensive and representative, and can be used to develop more robust object detection models that are capable of accurately identifying plastic waste in various lighting and weather conditions.

To capture the RGB images of the plastic waste, an iPhone 12 camera was used. The RGNIR images of plastic waste were captured using the Mapir Survey 3W, a multi-spectral surveying camera that is affordable, compact, and equipped with a wide-angle lens and high light sensitivity in the NIR range. The use of these two cameras allows for the acquisition of both RGB and RGNIR images, providing a diverse and representative dataset for evaluating the object detection model. The camera's sensitivity could capture 550 nm (green), 650 nm (red), and 850 nm (NIR) of light.

The two cameras were fixed on two separate heavy-duty tripod stands. Figure 4 shows the setup of the camera on the tripod in a real scene.

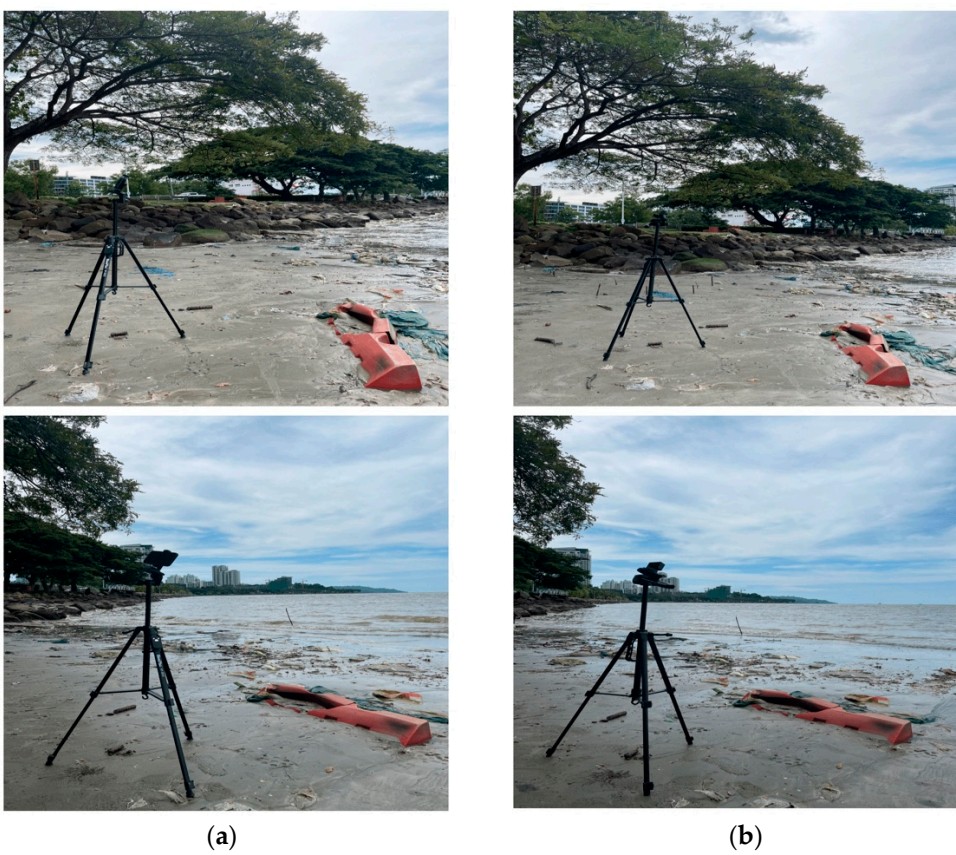

(**a**)                  (**b**)

**Figure 4.** The setup of the tripod with attached (**a**) iPhone 12 and (**b**) Mapir Survey 3W.

To ensure the acquisition of high-quality and representative images of plastic waste, the tripod stands were positioned at various distances and angles, with a focus on the plastic waste objects. There were two main criteria for setting the position of the tripod. The first criterion was that the distance of the camera from the plastic objects had to be adjusted at 1.0, 1.5, and 2.0 m apart. As a second criterion, the camera's angle, with respect to the subject, was set to the object's center, as well as 30 degrees to the left and right. By considering the angle of the camera in relation to the object, the dataset could be optimized to better represent the range of perspectives that the object detection model was expected to encounter in real-world scenarios. The setting of the camera used in this study was that for each distance, the camera was adjusted with three different angles. The image acquisition started off using the iPhone 12 to create the RGB datasets. Then, the RGB camera was swapped with the Mapir Survey 3W on a different tripod stand to create the RGNIR

datasets. As depicted in Figure 5, the position of the camera on the tripod was varied in terms of distance and angle to ensure the acquisition of a diverse and representative dataset for the evaluation of the object detection model.

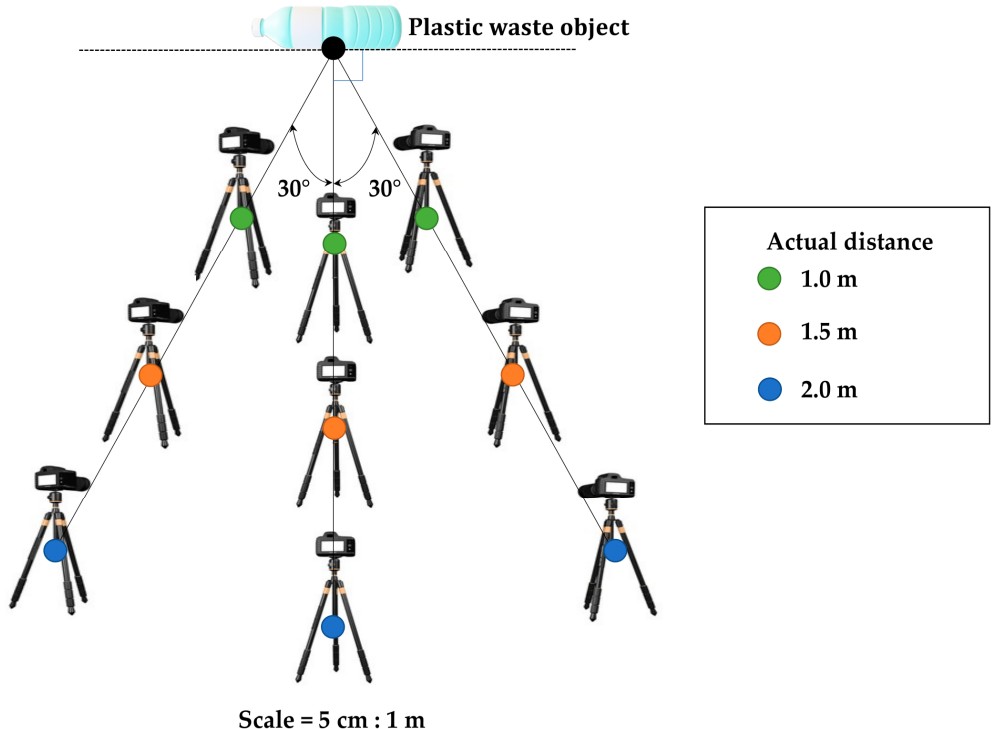

**Figure 5.** The tripod's position at various angles and distances.

A total of 45 scenes were selected as spots to position the tripod. A total of nine images with varying distances and angles were captured in every scene. Therefore, the overall number of plastic waste images for each RGB and RGNIR dataset was 405. Figures 6 and 7 show nine RGB and RGNIR images captured at the same scene.

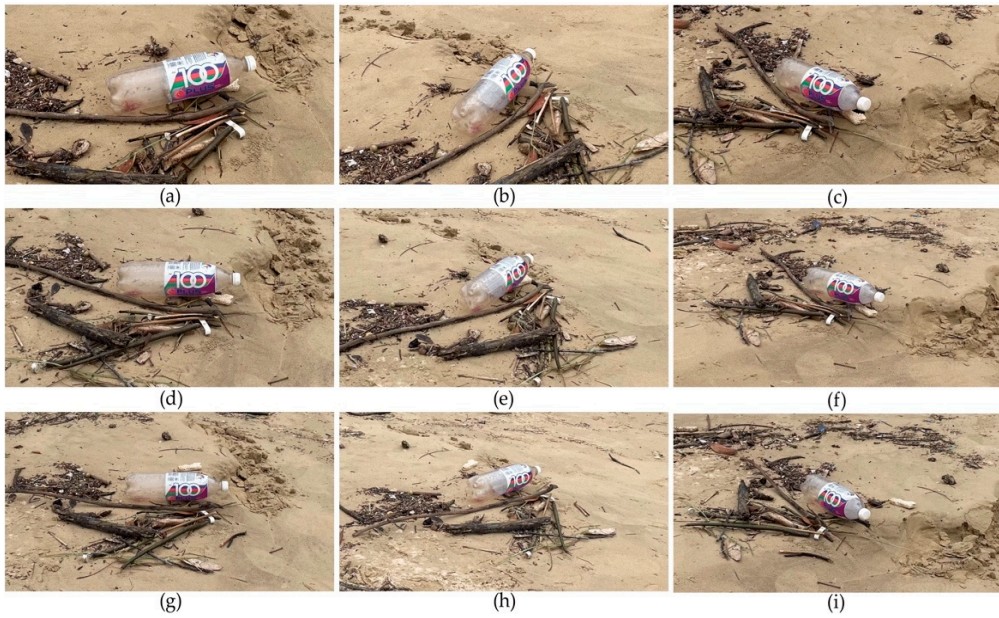

**Figure 6.** A sample of nine unique images of the plastic object from the RGB dataset; (**a**) 1.0 m and 30° left, (**b**) 1.0 m and center, (**c**) 1.0 m and 30° right, (**d**) 1.5 m and 30° left, (**e**) 1.5 m and center, (**f**) 1.5 m and 30° right, (**g**) 2.0 m and 30° left, (**h**) 2.0 m and center, and (**i**) 2.0 m and 30° right.

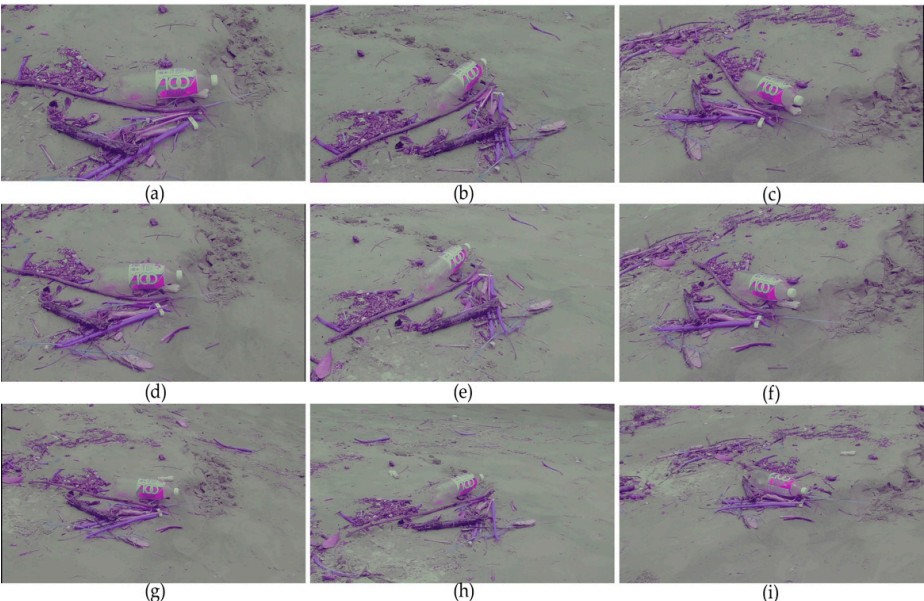

**Figure 7.** A sample of nine unique images of the plastic object from the RGNIR dataset; (**a**) 1.0 m and 30° left, (**b**) 1.0 m and center, (**c**) 1.0 m and 30° right, (**d**) 1.5 m and 30° left, (**e**) 1.5 m and center, (**f**) 1.5 m and 30° right, (**g**) 2.0 m and 30° left, (**h**) 2.0 m and center, and (**i**) 2.0 m and 30° right.

*4.2. Background Images*

The background images are images with no plastic objects present. It was reported that adding background images could decrease false positives during training [32]. To effectively reduce false positives, it was recommended that about 10% of the overall datasets be background images. The background images were taken from the exact three locations as described in Section 4.1. The settings for capturing the background images were the same as described in Section 4.1, in which each background image consisted of nine unique positions. Figures 8 and 9 show the samples of background images from RGB and RGNIR datasets, respectively.

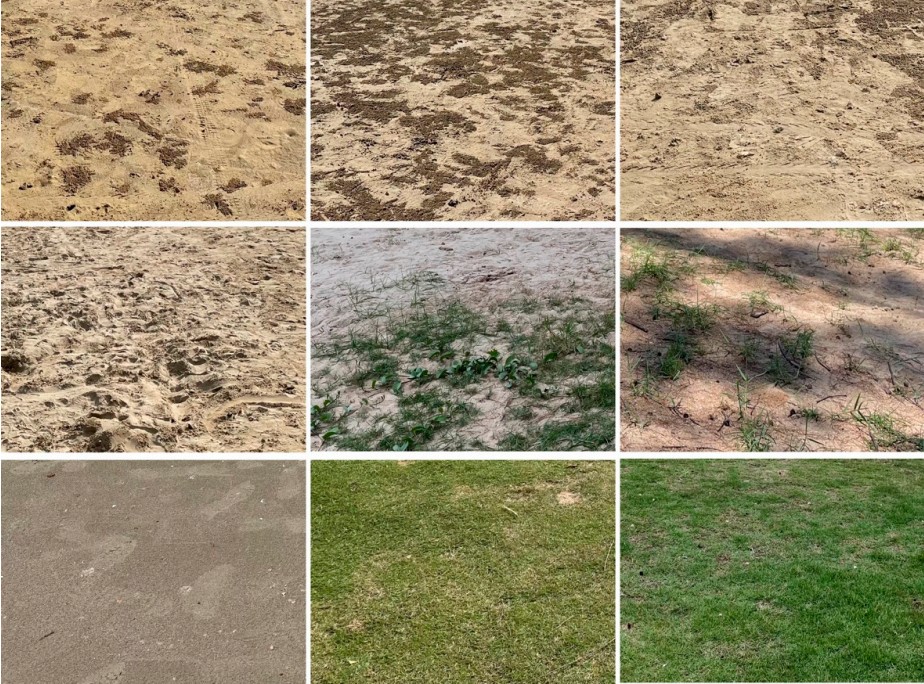

**Figure 8.** A sample of nine different images from the RGB dataset.

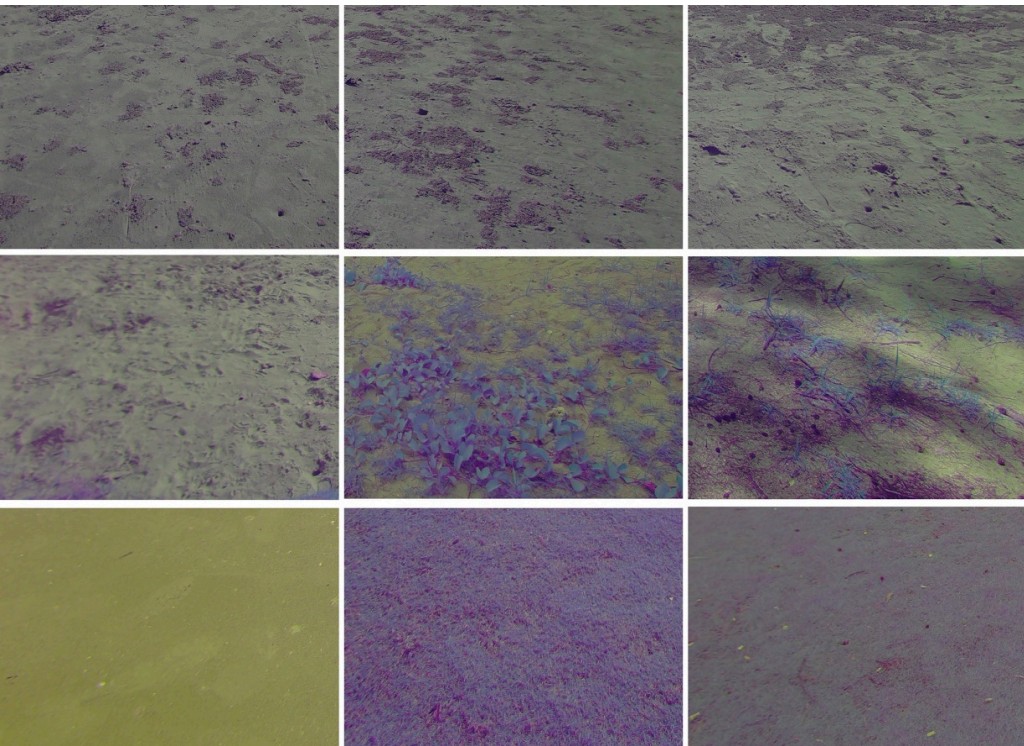

**Figure 9.** A sample of nine different images from the RGNIR dataset.

Before the acquired datasets were fed to the neural network, a series of pre-processing steps were applied to ensure the data was properly formatted and optimized for the object detection model. The specific pre-processing steps that were applied to the datasets are described in Section 4.3.

### 4.3. Pre-Processing

Initially, the images were cropped to eliminate any unrelated backgrounds, specifically non-plastics. Next, the images were resized, as the sizes of the images were not uniform after cropping. To be compatible with the YOLO model, which only accepts image sizes that are multiples of 32, the images were resized to a size of $416 \times 416$ pixels. This particular size was chosen as it effectively balanced the accuracy and speed of the YOLO model. A larger image size, exceeding $416 \times 416$, was not ideal as it consumed more computational time during processing. After that, the images underwent the labelling stage to draw ground truth boxes of the plastic object. An open-source graphical image annotation tool called LabelImg was used to label the images. This involved drawing bounding boxes around the objects in the images and labeling them. The annotations, which were saved as text files, are depicted in Figure 10. The use of LabelImg allows for the efficient and precise labeling of the images, a crucial aspect in the training and development of object detection models. As for the background images, annotation was not required. The background images were added directly to the training dataset. Figure 11 shows a sample of the overall flow of pre-processing steps on an image with multiple plastic waste objects. The distribution of the overall dataset with the annotations for RGB and RGNIR images is shown in Table 2.

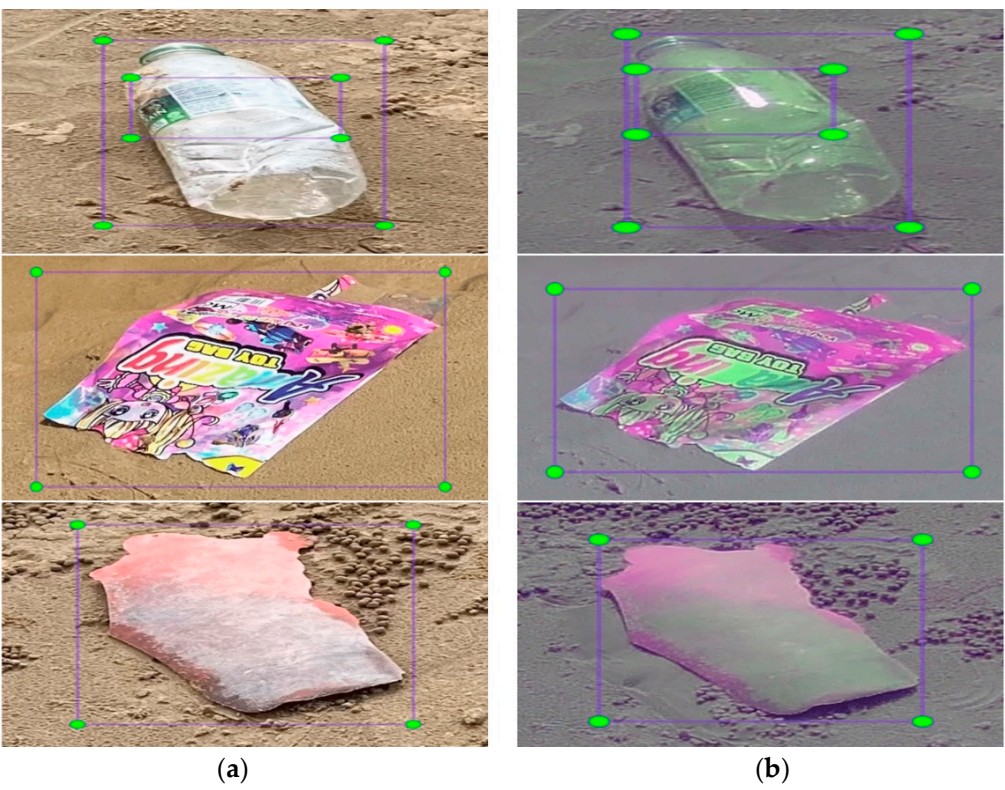

(**a**)  (**b**)

**Figure 10.** Annotation samples of (**a**) RGB and (**b**) RGNIR images.

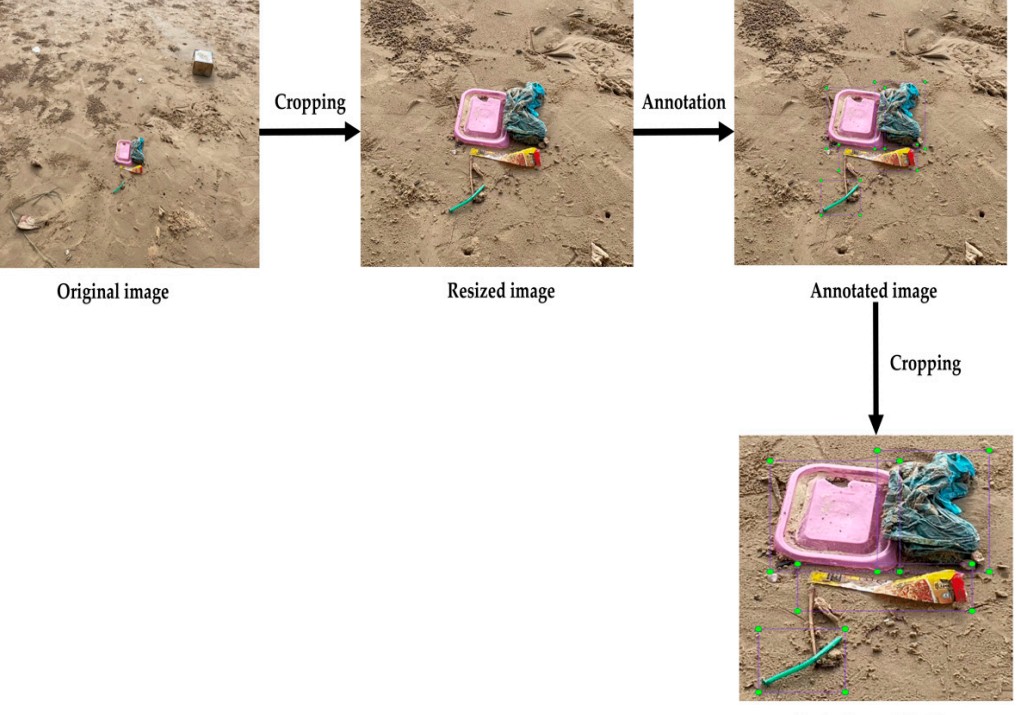

**Figure 11.** The sample of an overall flow of pre-processing steps for multiple plastic objects in an image.

**Table 2.** Distribution of plastic waste image dataset.

| Camera | Number of Images | | Number of Annotations |
|---|---|---|---|
| | Without Background | With Background | |
| iPhone 12 | 405 | 567 | 1344 |
| Mapir Survey 3W | 405 | 567 | 1344 |

*4.4. Dataset Partition*

The dataset was separated into three distinct sets following the pre-processing phase, including training, validation, and testing. The training set was used to allow the model to learn the key features present in the images, while the validation set was employed to validate the model's performance during training, providing valuable insights that could be used to optimize the model's hyperparameters and configurations for improved performance. The testing set, which consisted of unseen data, was used to evaluate the trained model's performance. In this paper, the ratio of training, validation, and testing was set as 70:20:10, as illustrated in Table 3. To ensure that the results could be replicated, the datasets were shuffled using the shuffle function of scikit-learn with the random-state parameter set to 1. This ensured that the data was randomly rearranged in a consistent manner.

**Table 3.** Distribution of plastic waste image dataset to training, validation, and testing dataset.

| Dataset | Number of Images | | | |
|---|---|---|---|---|
| | Training | | Validation | Testing |
| | Without Background | With Background | | |
| RGB | 284 | 446 | 81 | 40 |
| RGNIR | 284 | 446 | 81 | 40 |

**5. Experiments**

Figures 12 and 13 show the block diagrams of the first and second experiments, respectively, carried out in this work. The block diagram of the third experiment is shown in Figure 14. All plastic waste, including the background images, underwent pre-processing before the training, validation, and testing phases. First, all the plastic waste images were manually cropped to remove the non-plastic objects. Regarding the background images, any plastic objects that appeared in them were eliminated by cropping out the section of the image that contained plastics, resulting in a clean background. Then, the plastic waste images were annotated by providing the bounding boxes around the targeted objects as ground truth using LabelImg. The background images were not labeled and were simply added to the training dataset.

The datasets were then divided into three sets of experiments:

i. The first set of experiments consists of only plastic waste images. This experiment aims to compare the performance of the RGB and RGNIR datasets without the background images.

ii. The second set of experiments fuses the plastic waste images with the background images. This experiment aims to compare the performance of the RGB and RGNIR dataset with the background images. The background images are only added to the training dataset and are not used as part of the validation and testing dataset.

iii. The third experiment fuses both the RGB and RGNIR datasets to create more training samples. This experiment aims to investigate whether the model performs better in plastic waste detection that learns the RGB and RGNIR datasets features at the same time.

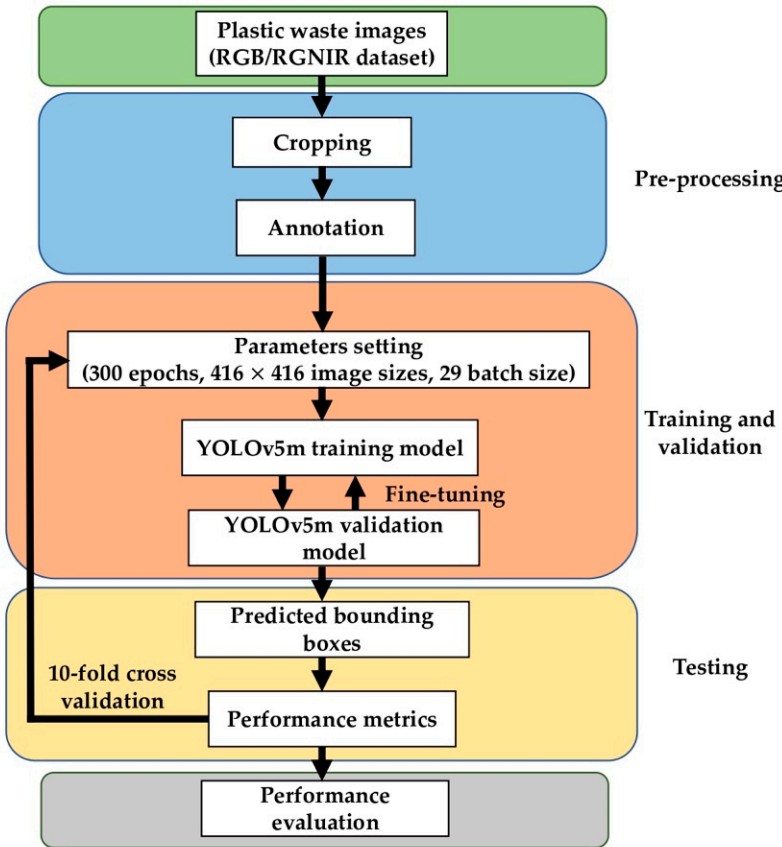

**Figure 12.** Flowchart of the first experiment.

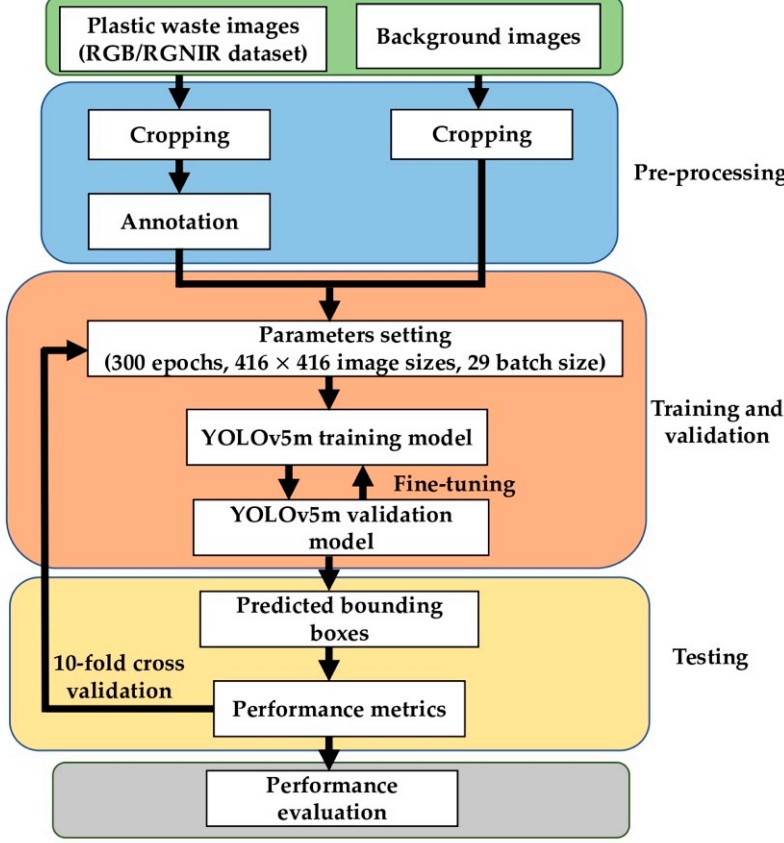

**Figure 13.** Flowchart of the second experiment.

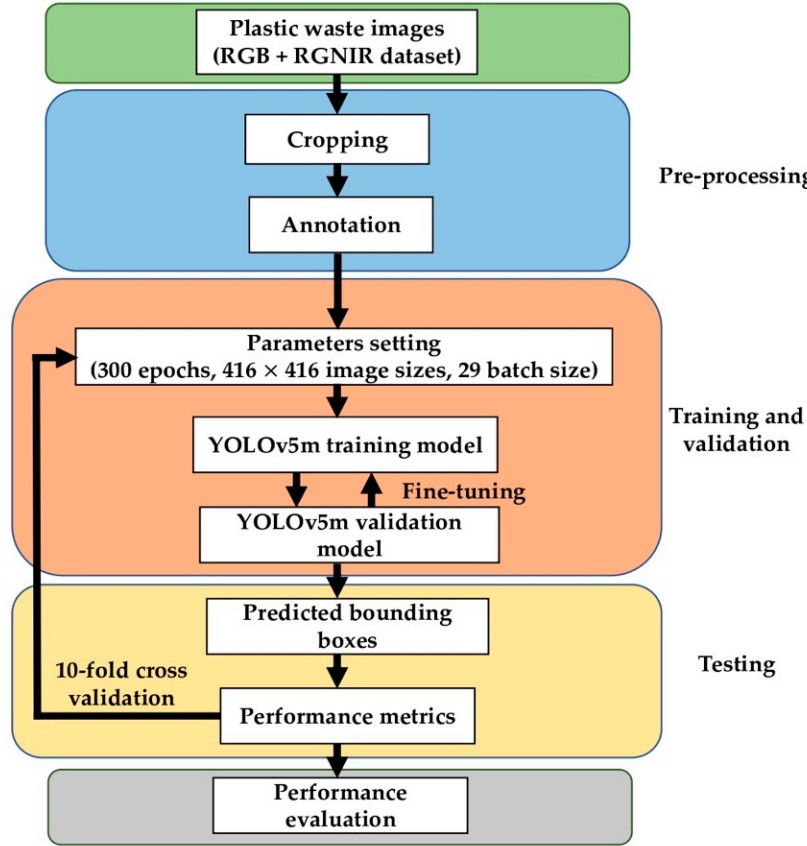

**Figure 14.** Flowchart of the third experiment.

All the experiments were carried out on the Jupyter Notebook as a training platform in Python for the YOLOv5m detection task. Pytorch 1.8.2, Intel (R) Core (TM) i7-10700, and the GPU of NVIDIA GeForce GTX 1660 Ti with CUDA 10.2 were used in this work. To ensure that all the training, validation, and testing datasets were evaluated on the proposed model, the cross-validation method was used. The cross-validation method is commonly utilized in the field of machine learning for evaluating the performance of a model on a limited data sample. A particular variant of cross-validation, known as *k*-fold cross-validation, involves partitioning the data into *k* groups, where the model is trained and validated on all but one group, and tested on the remaining group. In this paper, we utilized the *k*-fold cross-validation with *k* set to 10. This method involved dividing the data into ten subsets and using each subset in turn for testing and training the model on different iterations. While increasing the number of folds could lead to a reduction in bias and a more accurate estimate of the expected error, it also incurred a higher computational cost. In our case, we determined that 10-fold cross-validation offered the optimal balance between these two factors.

To ensure that every image in the dataset had an opportunity to be included in the training, validation, and testing phases, we shuffled the data and divided it into ten subsets. Seven subsets were used for training (70% of the data), two for validation (20%), and the remaining subset was reserved for testing (10%). This step is important in minimizing bias and reducing the risk of overfitting, ultimately allowing us to obtain a more accurate assessment of the performance of our proposed model.

## 6. Results and Discussion

In this section, we present the results of our performance evaluation and a comparison of the proposed model with state-of-the-art approaches. The performance metrics we used provides insight into the efficiency and effectiveness of our model in relation to the task at hand. By comparing our results to those of other leading approaches, we aim to

demonstrate the superiority of our proposed model and highlight its potential for practical application. Through this thorough analysis, we hope to contribute valuable insights to the field and further advance the state of the art in this area.

### 6.1. First Experiment: Training Dataset without Background Images

This subsection presents an in-depth evaluation of the performance of our proposed model on the datasets containing only plastic waste images. Using various performance metrics, we analyze the capabilities of the model in this specific context and provide a comprehensive discussion of the results.

### 6.1.1. Recall and Precision Results Using Training Dataset without Background Images

To optimize the performance of our neural network model, we conducted multiple iterations, known as epochs, through the entire dataset to fine-tune the weights of the network. Our analysis revealed that there was no significant improvement beyond 300 epochs for both the RGB and RGNIR datasets, leading us to conclude the training process for these datasets at this point. To ensure the most effective training possible, we carefully selected several key parameters for the model, including the input image size, batch size, initial and final learning rates, and weight decay. These values were chosen to maximize the model's ability to learn and generalize from the data, with an input image size of 416 × 416, a batch size of 32, an initial learning rate of 0.01, a final learning rate of 0.1, and a weight decay of 0.0005. As demonstrated in Figures 15 and 16, the YOLOv5m model is utilized to detect the validation and testing image samples, respectively. The proposed model accurately identified and localized plastic samples within the images by providing boundary boxes and confidence scores. The confidence score indicated the model's level of certainty that an object was present within a given bounding box and the accuracy of its estimation of the box based on the ground truth. These results demonstrate the impressive ability of the model to detect and classify plastic waste, highlighting its potential for use in practical scenarios.

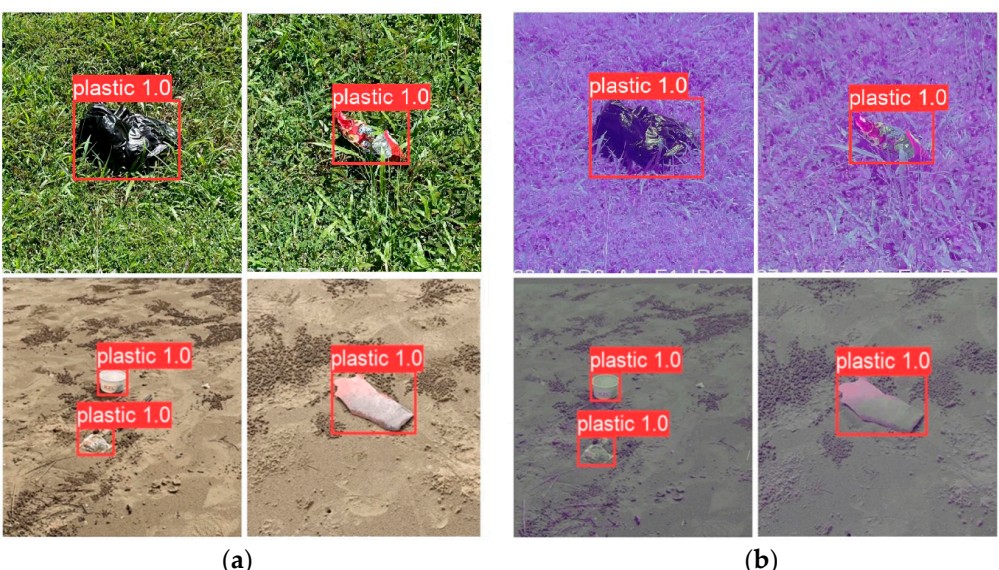

**Figure 15.** The YOLOv5m model was used to detect validation samples of (**a**) RGB images and (**b**) RGNIR images.

The proposed model underwent a rigorous evaluation process to measure its performance and reliability. Specifically, a 10-fold cross-validation approach was employed, utilizing the testing dataset as a means of evaluating the model's performance. The values shown in Table 4 are expressed as the mean ± standard deviation (SD), where SD is a measure of variation based on how far each data value differs from the mean. A high SD value indicates that the data values are more spread out, while a low SD value implies that the data values are more consistent, being clustered around the mean.

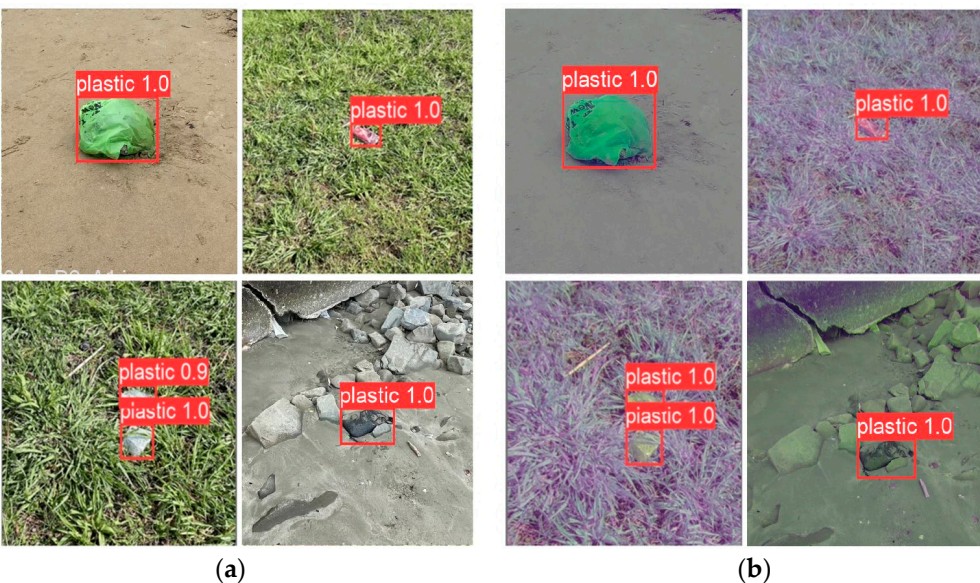

(**a**)　　　　　　　　　　　　　　　(**b**)

**Figure 16.** The YOLOv5m model was used to detect testing samples of (**a**) RGB images and (**b**) RGNIR images.

**Table 4.** Comparison of mean ± standard deviation of precision and recall metrics obtained through 10-fold cross-validation on a testing dataset, using a training dataset without background images for training.

| Training Dataset | Recall | Precision |
|---|---|---|
| RGB without Background Images | 90.52% ± 3.26% | 91.40% ± 2.86% |
| RGNIR without Background Images | 87.98% ± 4.00% | 93.91% ± 3.50% |

From Table 4, it can be seen that the RGB color space performed better than the RGNIR color space in terms of recall rate, whereas the RGNIR color space outperformed the RGB color space in terms of precision rate. This suggests that the RGB color space may be more suitable for detecting objects that look like plastic waste, but RGNIR is more precise in detecting the correct plastic waste object, at least in the context of the proposed model. At the same time, it can be observed that the standard deviation (SD) values indicate that the RGB color space is more stable than the RGNIR color space. This is evidenced by the lower SD values in the RGB as compared to the RGNIR.

6.1.2. Model mAP Results without Background Images in the Training Dataset

Table 5 presents the results of the proposed model's performance on the testing dataset using the mAP metric during the 10-fold cross-validation process. The mean mAP@0.5 values, which represent the mean mAP at an IoU threshold of 0.5, are 93.31% and 92.24% for the RGB and RGNIR datasets, respectively. Similarly, the mean mAP@0.5:0.95 values, which denote the mean mAP at IoU thresholds ranging from 0.5 to 0.95, are 67.99% and 69.23% for the RGB and RGNIR datasets, respectively. To further assess the model's performance, we calculated the Weighted Metric Score (WMS), which is based on the mean scores of mAP@0.5 and mAP@0.5:0.95 from the 10-fold cross-validation process. As shown in Equation (5), the WMS places greater weight (90%) on the more challenging and strict performance metric of mAP@0.5:0.95, with the remaining weight (10%) placed on mAP@0.5.

$$WMS = [(mAP@0.5) \times 0.1] + [(mAP@0.5:0.95) \times 0.9] \qquad (5)$$

**Table 5.** Testing dataset performance using training dataset without background images.

| Fold No. | RGB without Background Images | | | RGNIR without Background Images | | |
|---|---|---|---|---|---|---|
| | mAP@0.5 | mAP@0.5:0.95 | WMS | mAP@0.5 | mAP@0.5:0.95 | WMS |
| 1 | 90.32% | 63.59% | 66.27% | 90.08% | 68.65% | 70.79% |
| 2 | 95.22% | 71.23% | 73.63% | 94.33% | 70.49% | 72.87% |
| 3 | 93.58% | 67.94% | 70.50% | 90.94% | 65.99% | 68.48% |
| 4 | 97.61% | 70.51% | 73.22% | 98.80% | 77.62% | 79.74% |
| 5 | 92.88% | 68.19% | 70.66% | 90.65% | 64.65% | 67.25% |
| 6 | 94.59% | 72.65% | 74.84% | 93.31% | 72.48% | 74.57% |
| 7 | 90.83% | 62.78% | 65.59% | 86.94% | 63.31% | 65.67% |
| 8 | 95.47% | 68.42% | 71.12% | 93.08% | 69.67% | 72.01% |
| 9 | 91.36% | 68.33% | 70.63% | 92.09% | 68.88% | 71.20% |
| 10 | 91.24% | 66.28% | 68.77% | 92.16% | 70.60% | 72.75% |
| Mean | 93.31% | 67.99% | 70.52% | 92.24% | 69.23% | 71.53% |

From Table 5, the mean WMS for the RGB and RGNIR datasets was found to be 70.52% and 71.53%, respectively. A comparison of these values reveals that the performance of the RGNIR dataset was slightly superior to that of the RGB dataset, with a difference of approximately 1% based on the mean WMS. These results suggest that the RGNIR dataset may be more effective in detecting plastic waste using the proposed model, at least in the context of this study.

Table 6 presents a comparison of the mean mAP values of the RGB and RGNIR datasets, including their standard deviation. Based on the results presented in Table 6, regardless of the performance metric, it is evident that the RGB consistently demonstrates a lower STD value compared to the RGNIR. However, the outcomes reported in the WMS column suggest that the proposed model may be more effective in identifying plastic waste using the RGNIR dataset. This is further supported by the RGNIR's WMS results, which are about 1% higher than those obtained using the RGB. The following two conclusions can be drawn: (i) if consistency is the priority, then RGB may be considered as the preferred color space; (ii) if higher mAP performance is the priority, then RGNIR may be selected as the preferred color space. The higher average performance of the model on the RGNIR dataset, compared to the RGB dataset, suggests that the RGNIR images are more suitable for object detection models in the context of detecting plastic waste using visible light spectrum data. It is important to note that these findings are based on the 10-fold cross-validation process and may not be generalizable to other contexts or datasets. Nonetheless, these insights provide valuable information regarding the performance of the proposed model and the potential advantages of using certain datasets or color spaces for detecting plastic waste.

**Table 6.** Comparison of mean $\pm$ standard deviation of mAP with standard deviation obtained through 10-fold cross-validation on a testing dataset, using a training dataset without background images for training.

| Training Dataset | mAP@0.5 | mAP@0.5:0.95 | WMS |
|---|---|---|---|
| RGB without Background Images | 93.31% $\pm$ 2.28% | 67.99% $\pm$ 2.97% | 70.52% $\pm$ 2.86% |
| RGNIR without Background Images | 92.24% $\pm$ 2.94% | 69.23% $\pm$ 3.89% | 71.53% $\pm$ 3.78% |

*6.2. Second Experiment: Training Dataset with Background Images*

This subsection discusses the results of the proposed model's performance on the datasets that contain both plastic waste and background images.

6.2.1. Recall and Precision Results Using Training Dataset with Background Images

Table 7 shows the recall and precision rates for the testing dataset obtained through the 10-fold cross-validation process with background images. Overall, it was found that the mean values of the recall and precision did not differ significantly for the RGB and RGNIR datasets with background images. Regardless of the performance metric, RGB always outperforms RGNIR, with its STD consistently being lower than RGNIR. If background images that are free of plastic waste are included in the training dataset, it can be concluded

that the RGB dataset is more effective in identifying targets compared to the RGNIR dataset. Therefore, in applications where accurate target detection is required, the use of RGB imagery may be preferred.

**Table 7.** Comparison of mean ± standard deviation of precision and recall metrics obtained through 10-fold cross-validation on a testing dataset, using a training dataset with background images for training.

| Training Dataset | Recall | Precision |
|---|---|---|
| RGB with Background Images | 89.17% ± 2.75% | 93.51% ± 2.42% |
| RGNIR with Background Images | 88.64% ± 2.95% | 93.05% ± 3.17% |

6.2.2. Model mAP Results with Background Images in the Training Dataset

Tables 8 and 9 show that using the RGB dataset with background images in the training dataset results in better performance on the mAP@0.5 metric. However, the RGNIR dataset with background images performs slightly better than the RGB dataset on the mAP@0.5:0.95 and WMS metrics. These results are similar to those presented in Tables 5 and 6, where the dataset that uses RGB color space performed better on the mAP@0.5 metric, while the dataset that uses RGNIR color space performed better on the mAP@0.5:0.95 and WMS metrics. Similar to the results presented in Table 6, the results presented in Table 9 show that the RGB dataset without background images generally gives lower STD regardless of the performance metrics. It is worth considering that the inclusion of background images in the training dataset may have influenced the model's performance, as the presence of background clutter can impact the accuracy of object detection models. Therefore, careful consideration should be given to the choice of the dataset and the inclusion or exclusion of background images when developing and evaluating object detection models. Further research may be necessary to fully understand the role that background images play in the model's performance and to identify any potential biases in the training data.

**Table 8.** Testing dataset performance using training dataset with background images.

| Fold No. | RGB with Background Images | | | RGNIR with Background Images | | |
|---|---|---|---|---|---|---|
| | mAP@0.5 | mAP@0.5:0.95 | WMS | mAP@0.5 | mAP@0.5:0.95 | WMS |
| 1 | 90.47% | 63.69% | 66.36% | 90.36% | 68.30% | 70.51% |
| 2 | 95.49% | 71.10% | 73.54% | 94.71% | 71.24% | 73.59% |
| 3 | 92.99% | 66.67% | 69.30% | 90.38% | 65.40% | 67.90% |
| 4 | 98.29% | 69.99% | 72.82% | 97.82% | 77.99% | 79.98% |
| 5 | 92.37% | 66.83% | 69.39% | 90.79% | 65.31% | 67.85% |
| 6 | 94.27% | 71.98% | 74.21% | 94.42% | 70.90% | 73.25% |
| 7 | 90.30% | 63.07% | 65.79% | 87.04% | 62.61% | 65.06% |
| 8 | 95.90% | 68.18% | 70.95% | 92.34% | 66.96% | 69.50% |
| 9 | 93.01% | 68.57% | 71.01% | 92.34% | 70.66% | 72.83% |
| 10 | 90.31% | 66.50% | 68.88% | 92.51% | 70.87% | 73.03% |
| Mean | 93.34% | 67.66% | 70.23% | 92.27% | 69.02% | 71.35% |

**Table 9.** Comparison of mean ± standard deviation of mAP with standard deviation obtained through 10-fold cross-validation on a testing dataset, using a training dataset with background images for training.

| Training Dataset | mAP@0.5 | mAP@0.5:0.95 | WMS |
|---|---|---|---|
| RGB with Background Images | 93.34% ± 2.54% | 67.66% ± 2.77% | 70.23% ± 2.70% |
| RGNIR with Background Images | 92.27% ± 2.80% | 69.02% ± 4.10% | 71.35% ± 3.95% |

*6.3. Third Experiment: Training Dataset That Consists of Images from a Fused RGB and RGNIR Datasets*

In this subsection, we present and analyze the performance of the proposed model using a fusion of RGB and RGNIR datasets that contained a greater number of images of plastic waste. The purpose of fusing the RGB and RGNIR datasets in the training dataset

was to investigate whether the fused dataset could cause the model to learn the characteristics and features of plastic waste better, potentially leading to improved performance.

6.3.1. Recall and Precision Results from Training on Fused RGB and RGNIR Image Datasets

The performance of the proposed model on the fused RGB and RGNIR training datasets in terms of recall and precision rates from the 10-fold cross-validation process is presented in Table 10. As shown in Table 10, it can be observed that the recall and precision results of the fused RGB and RGNIR training datasets are satisfactory, achieving a mean recall and precision rate of 89.32% ± 3.79% and 93.06% ± 1.87%, respectively.

**Table 10.** Comparison of mean ± standard deviation of precision and recall metrics obtained through 10-fold cross-validation on a testing dataset, using a fused RGB and RGNIR training dataset.

| Training Dataset | Recall | Precision |
| --- | --- | --- |
| Fusion of RGB and RGNIR Dataset | 89.32% ± 3.79% | 93.06% ± 1.87% |

6.3.2. Model mAP Results from Training on Fused RGB and RGNIR Image Datasets

Tables 11 and 12 present the results of the proposed model's mAP on the testing dataset from the 10-fold cross-validation process. The results demonstrate that the fusion of the RGB and RGNIR datasets during testing achieved a mean mAP@0.5 of 92.96%, as well as a mean mAP@0.5:0.95 of 69.47% and a mean WMS of 71.82%. To get a better sense of which experimental criteria give the best outcomes in terms of performance metrics, Table 13 was prepared to present the overall generalizability of the model on different sets of experiments (Experiment One, Two, and Three) and to identify which settings generate the best generalizability results.

**Table 11.** Testing dataset performance using fused RGB and RGNIR training dataset.

| Fold No. | Fusion of RGB and RGNIR Dataset | | |
| --- | --- | --- | --- |
| | mAP@0.5 | mAP@0.5:0.95 | WMS |
| 1 | 90.84% | 65.87% | 68.37% |
| 2 | 95.52% | 72.41% | 74.72% |
| 3 | 92.68% | 68.49% | 70.91% |
| 4 | 98.07% | 74.40% | 76.77% |
| 5 | 91.53% | 67.24% | 69.67% |
| 6 | 94.68% | 73.97% | 76.04% |
| 7 | 88.55% | 64.50% | 66.91% |
| 8 | 94.32% | 69.32% | 71.82% |
| 9 | 92.89% | 69.17% | 71.54% |
| 10 | 90.54% | 69.34% | 71.46% |
| Mean | 92.96% | 69.47% | 71.82% |

**Table 12.** Comparison of mean ± standard deviation of mAP with standard deviation obtained through 10-fold cross-validation on a testing dataset, using a fused RGB and RGNIR training dataset.

| Training Dataset | mAP@0.5 | mAP@0.5:0.95 | WMS |
| --- | --- | --- | --- |
| Fusion of RGB and RGNIR Dataset | 92.96% ± 2.63% | 69.47% ± 3.11% | 71.82% ± 3.04% |

Table 13 compares three sets of experiments based on recall, precision, mAP@0.5, mAP@0.5:0.95, and WMS obtained from the 10-fold cross-validation process. The best mean recall rate of 90.52% ± 3.26% is reported by the RGB training dataset without background images. The best mean precision rate of 93.91% ± 3.50% is reported by the RGNIR training dataset without background images. The best mean mAP@0.5 rate of 93.34% ± 2.54% is reported by the RGB training dataset with background images. The

best mean mAP@0.5:0.95 rate of 69.47% $\pm$ 3.11% is reported by the fusion of the RGB and RGNIR training datasets. Finally, the best mean WMS of 71.82% $\pm$ 3.04% is reported by the fusion of the RGN and RGNIR training datasets. One critical implication is that the choice of the training dataset can significantly impact the performance of the model in object detection tasks. Specifically, the best-performing dataset varies depending on the performance metric of interest. This suggests that researchers and practitioners must carefully consider the selection of training datasets for their object detection models to achieve optimal performance. Additionally, the results demonstrate that the fusion of different datasets can improve overall performance, highlighting the potential benefits of using multiple sources of training data in object detection applications. These findings emphasize the importance of conducting rigorous experiments and evaluations to identify the most effective training datasets and techniques for object detection models.

**Table 13.** A comparison mean $\pm$ standard deviation of precision, recall, and WMS using testing dataset from 10-fold cross-validation.

| Experiment | Dataset | Recall | Precision | mAP@0.5 | mAP@0.5:0.95 | WMS |
|---|---|---|---|---|---|---|
| First experiment | RGB without background images | 90.52% $\pm$ 3.26% | 91.40% $\pm$ 2.86% | 93.31% $\pm$ 2.28% | 67.99% $\pm$ 2.97% | 70.52% $\pm$ 2.86% |
| | RGNIR without background images | 87.98% $\pm$ 4.00% | 93.91% $\pm$ 3.50% | 92.24% $\pm$ 2.94% | 69.23% $\pm$ 3.89% | 71.53% $\pm$ 3.78% |
| Second experiment with background images | RGB with background images | 89.17% $\pm$ 2.75% | 93.51% $\pm$ 2.42% | 93.34% $\pm$ 2.54% | 67.66% $\pm$ 2.77% | 70.23% $\pm$ 2.70% |
| | RGNIR with background images | 88.64% $\pm$ 2.95% | 93.05% $\pm$ 3.17% | 92.27% $\pm$ 2.80% | 69.02% $\pm$ 4.10% | 71.35% $\pm$ 3.95% |
| Third experiment with a fusion of RGB and RGNIR datasets | Fusion of RGB and RGNIR dataset | 89.32% $\pm$ 3.79% | 93.06% $\pm$ 1.87% | 92.96% $\pm$ 2.63% | 69.47% $\pm$ 3.11% | 71.82% $\pm$ 3.04% |

Typically, one aims for a balance between precision and recall, and the mAP@0.5:0.95 metric is commonly used to assess the overall performance in object detection tasks. In the third experiment, the proposed model displayed the best overall mean performance, achieving the highest mAP@0.5:0.95 score of 69.47% $\pm$ 3.11%. This may have been due, in part, to the inclusion of both the visible light spectrum and the NIR information in the training dataset, as well as the increase in the number of training images when fusing both RGB and RGNIR datasets. It was well-established that the object detection models tended to perform better as the size of the training dataset increased, as the model could learn a greater variety of features and patterns. Therefore, the increased size of the training dataset in the third experiment likely contributed to the improved performance of the proposed model. Overall, these results highlighted the importance of carefully selecting and pre-processing training data to achieve optimal performance in object detection tasks.

To further summarize Table 13, it appears that the RGNIR color space performs similarly to the RGB in terms of recall and precision. However, the RGNIR appears to have slightly higher precision in most cases, suggesting that it may be more accurate in detecting objects than the RGB. When it comes to mAP@0.5:0.95, the fusion of the RGB and RGNIR datasets outperformed the individual RGB and RGNIR datasets. This indicates that the combination of both color spaces can provide a more comprehensive and accurate representation of objects, leading to improved object detection performance. The inclusion of NIR information in the training dataset can potentially provide additional spectral information that may be useful for object detection tasks. NIR light, which has longer wavelengths than visible light, can penetrate certain materials and provide information about the internal structure of objects, making it useful for detecting and localizing plastic waste. However, using NIR imaging in object detection tasks requires careful consideration of its potential implications, including atmospheric conditions, surface reflectance, and the presence of other materials that can affect image quality and reliability. Moreover, NIR information may not always be relevant or necessary for certain object detection tasks,

and its inclusion in the training dataset may introduce unnecessary complexity and bias. Therefore, thorough evaluation and analysis are essential to fully comprehend the role of NIR information in the performance of object detection models, as well as its potential benefits and limitations.

### 6.4. Fourth Experiment: A Comparison of Faster R-CNN and YOLOv5m with Fused RGB and RGNIR Image Dataset

In this paper, a comparison between Faster R-CNN and YOLOv5 is necessary to evaluate the performance of these two algorithms for object detection. While previous studies have shown that YOLOv5 outperforms other state-of-the-art object detection algorithms in terms of speed and accuracy, including Faster R-CNN, it is still important to assess how well these algorithms perform when compared to each other in specific scenarios. The choice of Faster R-CNN and YOLOv5 as the algorithms for comparison is relevant because they represent two different approaches to object detection. Faster R-CNN is based on region proposals and uses a convolutional neural network to detect and classify objects, while YOLOv5 is a single-stage object detection algorithm that directly predicts object bounding boxes and class probabilities in a single pass through the network.

#### 6.4.1. Faster R-CNN as Object Detection Model

Faster Region-Based Convolutional Neural Network (R-CNN) is a deep learning-based algorithm used for object detection in computer vision. It has gone through a development process from R-CNN to Fast R-CNN, and finally to Faster R-CNN. Faster R-CNN is an object detection algorithm that combines the strengths of R-CNN and Fast R-CNN. It uses a Region Proposal Network (RPN) to generate object proposals directly from the convolutional feature map, eliminating the need for an external proposal method. The RPN generates object proposals by sliding a small network over the convolutional feature map and predicting whether an object is present at each sliding window location. Once the object proposals are generated, they are passed to the RoI pooling layer, which extracts features from each proposal and feeds them into a fully connected network for classification and bounding box regression [33]. Figure 17 illustrates the Faster R-CNN structure.

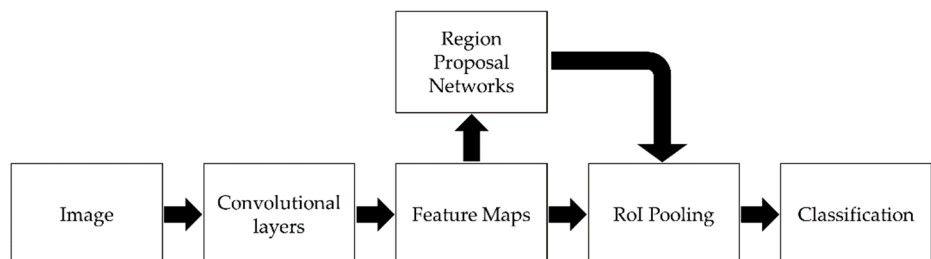

**Figure 17.** Faster R-CNN structure.

In this paper, ResNet50 was selected as the backbone for Faster R-CNN over ResNet101 due to its superior computational efficiency and shorter training times, despite the potential for ResNet101 to achieve higher accuracy. This decision was made to ensure a fair comparison between the Faster R-CNN and the YOLOv5 architecture. Specifically, a Faster R-CNN model with ResNet50 as the backbone, which is less complex than ResNet101, is compared to YOLOv5m, which is representative of the YOLOv5 family, but not the most complex model in that family.

#### 6.4.2. Faster R-CNN and YOLOv5m mAP Results from Training on Fused RGB and RGNIR Image Datasets

Table 14 presents the results of Faster R-CNN and YOLOv5m mAP on the same testing dataset from the 10-fold cross-validation process.

**Table 14.** A comparison mean ± standard deviation of Faster R-CNN and YOLOv5m mAP@0.5, mAP@0.5:0.95, and WMS using testing dataset from 10-fold cross-validation.

| Experiment | mAP@0.5 | mAP@0.5:0.95 | WMS |
|---|---|---|---|
| Faster R-CNN (ResNet50 as backbone) | 71.34% ± 3.93% | 44.05% ± 3.36% | 46.78% ± 3.39% |
| YOLOV5m | 92.96% ± 2.63% | 69.47% ± 3.11% | 71.82% ± 3.04% |

The results in Table 14 show that YOLOv5m outperforms Faster R-CNN in all performance metrics. YOLOv5m achieves a significantly higher mAP@0.5 rate of 92.96% ± 2.63% compared to Faster R-CNN's rate of 71.34% ± 3.93%. Similarly, YOLOv5m achieves a higher mAP@0.5:0.95 rate of 69.47% ± 3.11% compared to Faster R-CNN's rate of 44.05% ± 3.36%. Finally, YOLOv5m achieves a higher WMS of 71.82% ± 3.04% compared to Faster R-CNN's WMS of 46.78% ± 3.39%. These results suggest that YOLOv5m has superior performance compared to Faster R-CNN when it comes to object detection accuracy.

*6.5. Comparison with State-of-the-Art Approaches*

Table 15 presents a comparison of the proposed YOLOv5m model with other state-of-the-art YOLOv5 approaches that were applied in recent studies on waste detection. It is important to note that the studies used for comparison were carefully selected to ensure a fair comparison, with all of them containing at least plastic waste in their datasets. It is also worth noting that the other five state-of-the-art approaches utilized RGB datasets as part of their training images, while the proposed approach utilized both RGB and RGNIR datasets containing visible light and NIR images for fitting and testing evaluation. Overall, the comparison presented in Table 15 highlights the potential benefits of using both RGB and NIR information in object detection tasks, particularly for the detection of plastic waste. Further research may be necessary to fully understand the role that NIR information plays in the performance of object detection models and to identify any potential biases in the training data.

**Table 15.** Comparison between the proposed model with other approaches.

| Method | Color Space | mAP@0.5 | mAP@0.5:0.95 | Waste Type |
|---|---|---|---|---|
| GC-YOLOv5 [25] | RGB | 99.59% | 64.70% | Electronics, fruits, papers, and plastics |
| YOLOv5-Attention-KG [34] | RGB | 73.20% | - | Recyclable, food, and hazardous |
| YOLOv5s-CSS [26] | RGB | 98.30% | - | Plastics |
| FMA-YOLOv5s [27] | RGB | 88.54% | - | Plastics |
| YOLOv5s [28] | RGB | 85.00% | - | Plastics |
| YOLOv5m (Proposed model) | RGB without background images | 93.31% ± 2.28% | 67.99% ± 2.97% | Plastics |
| YOLOv5m (Proposed model) | RGNIR without background images | 92.24% ± 2.94% | 69.23% ± 3.89% | Plastics |
| YOLOv5m (Proposed model) | RGB with background images | 93.34% ± 2.54% | 67.66% ± 2.77% | Plastics |
| YOLOv5m (Proposed model) | RGNIR with background images | 92.27% ± 2.80% | 69.02% ± 4.10% | Plastics |
| YOLOv5m (Proposed model) | Fusion of RGB and RGNIR dataset | 92.96% ± 2.63% | 69.47% ± 3.11% | Plastics |

The results presented in Table 15 suggest that the proposed model, which utilizes a fusion of RGB and NIR information in the training dataset, outperforms some of the other state-of-the-art YOLOv5 approaches in terms of mAP@0.5, while performing worse than others. Nevertheless, the proposed model demonstrates the greatest mAP@0.5:0.95 value

compared to all other approaches, indicating its potential to perform better if both visible light spectrum and NIR spectrum data are used as feature representation rather than a single color space. This finding suggests the need for further investigation to explore the benefits of using both types of information in object detection tasks, particularly for the detection and localization of plastic waste. The potential advantage of the proposed model is its ability to use an 8-bit NIR-spectrum representation acquired by a camera, which makes it practical for outdoor use and adds to its feasibility for real-world applications. Despite these advantages, the performance of the proposed model in detecting plastic waste could be further improved by considering the ability of NIR light to penetrate certain materials and provide information about the internal structure of objects. Overall, the findings highlight the potential benefits of utilizing both RGB and NIR information in object detection tasks, but further research is necessary to fully understand the role of NIR information in improving object detection performance.

## 7. Conclusions

In conclusion, the results presented in this research paper indicate that the choice of the training dataset can significantly impact the performance of object detection models. Specifically, the best-performing dataset varies depending on the performance metric of interest. The fusion of different datasets can improve overall performance, highlighting the potential benefits of using multiple sources of training data in object detection applications. These findings emphasize the importance of conducting rigorous experiments and evaluations to identify the most effective training datasets and techniques for object detection models. Moreover, the proposed model exhibited the best overall mean performance in the third experiment, generating the highest mAP@0.5:0.95 score of 69.47% $\pm$ 3.11%. This may have been due to the inclusion of both the visible light spectrum and NIR information in the training dataset, as well as the increase in the number of training images when fusing both the RGB and RGNIR datasets. It was well established that object detection models tended to perform better as the size of the training dataset increased, as the model could learn a greater variety of features and patterns. Therefore, the increased size of the training dataset in the third experiment likely contributed to the improved performance of the proposed model. The inclusion of NIR information in the training dataset can potentially provide additional spectral information that may be useful for object detection tasks, especially for detecting and localizing plastic waste. However, the use of NIR imaging in object detection tasks requires careful consideration of its potential implications, including atmospheric conditions, surface reflectance, and the presence of other materials that can affect image quality and reliability. Moreover, NIR information may not always be relevant or necessary for certain object detection tasks, and its inclusion in the training dataset may introduce unnecessary complexity and bias. In summary, this research highlights the importance of carefully selecting and pre-processing training data to achieve optimal performance in object detection tasks. Future work could explore the use of other color spaces and feature engineering techniques to further improve object detection performance. Additionally, further research is needed to understand the potential benefits and limitations of using NIR information in object detection tasks and to develop robust methods for incorporating it into training datasets.

## 8. Future Works

This study utilized the medium YOLOv5 architecture (specifically the YOLOV5m model) to evaluate the proposed datasets, while acknowledging that other model sizes were not explored. Future research should evaluate these alternative architectures, as well as increase the number and diversity of images used to enhance the model's performance, explore extending the spectral range beyond 850 nm, and test the robustness of the model under different conditions. Other potential avenues for future research include testing the robustness of the proposed model at different distances from the camera. Additionally, the study proposed examining the impact of different image resolutions on the model's

performance and using higher GPU specifications to reduce training time. In addition to the future work mentioned, further feature engineering can be explored to enhance the performance of object detection models. One approach is to incorporate advanced feature extraction techniques such as edge detection, texture analysis, and shape analysis. These features can provide additional information about the shape and texture of objects, which can improve the accuracy of object detection algorithms. Another potential feature engineering approach is to explore the use of semantic segmentation to improve the model's ability to detect and classify objects in complex scenes. Furthermore, the use of data augmentation techniques can help to increase the diversity of the training data, improving the model's ability to generalize to new data. Data augmentation can include techniques such as rotation, scaling, flipping, and cropping, which can create additional variations of the training images.

**Author Contributions:** Conceptualization, O.T., E.G.M. and J.A.D.; data curation, E.G.M. and O.T.; formal analysis, E.G.M., O.T., F.Y. and A.F.; funding acquisition, E.G.M.; investigation, E.G.M. and O.T.; methodology, E.G.M., O.T. and J.A.D.; writing—review and editing, E.G.M., L.A., F.Y., F.S. and N.F.M.N.; project administration, E.G.M. and J.A.D.; resources, E.G.M.; supervision, E.G.M. and J.A.D. All authors have read and agreed to the published version of the manuscript.

**Funding:** This research is supported by the Ministry of Higher Education Malaysia through the Fundamental Research Grant Scheme (FRGS/1/2020/ICT06/UMS/02/1).

**Institutional Review Board Statement:** Not applicable.

**Informed Consent Statement:** Not applicable.

**Data Availability Statement:** The dataset used in our study can be accessed upon reasonable request through the corresponding authors.

**Acknowledgments:** The authors thank the Ministry of Higher Education Malaysia and Universiti Malaysia Sabah for supporting this study.

**Conflicts of Interest:** The authors declare no conflict of interest.

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
