# Peer review of "On-Shore Plastic Waste Detection with YOLOv5 and RGB-Near-Infrared Fusion: A State-of-the-Art Solution for Accurate and Efficient Environmental Monitoring"

_2504-2289, doi:10.3390/bdcc7020103_

Round 1

Reviewer 1 Report

The article respects the rigors that a scientific paper must do it.

This is topical issue.

A professional analysis, correlated with the results of the investigation, the calculation and with the representations of  well done tables, graphs and photos are   made by the  authors.

Abstract - it  is  a self-contained unit capable of being understood without the benefit of the others chapters.

 Introduction - Section 1 well provides the background information necessary to understand  the topic. Related Works- Section 2 The literature reviewed is systematic and well done and  is related with YOLOv5s architecture.. All eight sections of the article create a unitary whole that argues and describes the theme of the work.

all eight sections of the article create a unitary whole that argues and describes the theme of the work.

Reviewer 2 Report

I really liked this MS and there were few mistakes in the text. 

1. My main comment which might take a bit of work is that scientific studies should be published in the past tense. Where this paper is all the present text. 

2. Are all the tables of results necessary. I feel some of these could be dropped or put into supplementary. 

The authors did not overstate the results and were open about the limitation of the work. 

Reviewer 3 Report

The manuscript title "On-shore Plastic Waste Detection with YOLOv5 and RGB-  Near-Infrared Fusion: A State-of-the-Art Solution for Accurate and Efficient Environmental Monitoring" is well designed and presented. Although, I have few major concerns:

1) The plastic waste problems are not well introduced in Abstract. 
2) Similarly, the problems related to plastic waste detection are not highlighted properly, justification is required.
3) A tabular description could be added including dataset, year, method and results in related work section.
4) The methodology needs justification about why YOLOV5 is used to implemented, why not other methods?
5) The features engineering could be added to improve the results. 
6) A comparison is also needed to proven the validity. 
7) Improve the conclusion part with addition to your study weakness, strength and future work. 

Reviewer 4 Report

In the study utilization of data from 2 spectra for plastic waste detection was examined.

For this purpose YOLOv5 based system applied for RGB and RGNIR photographs was used.

Detailed description of methods and data analysis is provided.

Although the findings are not groundbreaking, they make a valuable contribution to data processing techniques for plastic detection and localization.

Q. 1. Were daylights condition related to day time or sunny/cloudy weather taken into account in the studies?

Some corrections in text are required

l. 129 splitting the word

l. 150 explain “[email protected], and [email protected]:0.95”. Besides Abstract it is the first occurence of acronyms

l. 570 double dot

Round 2

Reviewer 3 Report

Thanks for revising manuscript according to my major concerns. The revised manuscript is well in well improved form. However, as you added justification about YOLOv5 in line 238-244. Please add the performance on your dataset of other methods to prove your statement about YOLOV5 performance. 

Author Response

Dear Reviewer,

Thank you for providing us with your insightful feedback, which has significantly contributed to improving the quality of our paper. In response to your comments, we have included additional content in section 6.4, specifically in lines 699 to 734. In this section, we have presented a comparison of Faster R-CNN and YOLOv5m using the proposed dataset. We have trained and evaluated Faster R-CNN on the dataset and subsequently compared its performance with that of YOLOv5m using the same test bed.

Thank you again for your feedback and suggestions, which have helped us to enhance the quality and rigor of our paper.
